# Germanium-Tin (GeSn) Metal-Semiconductor-Metal (MSM) Near-Infrared Photodetectors

**DOI:** 10.3390/mi13101733

**Published:** 2022-10-14

**Authors:** Ricky Wenkuei Chuang, Yu-Hsin Huang, Tsung-Han Tsai

**Affiliations:** Institute of Microelectronics, Department of Electrical Engineering, National Cheng Kung University, Tainan 70101, Taiwan

**Keywords:** metal–semiconductor–metal (MSM) photodetectors, germanium–tin (GeSn), near-infrared, distributed Bragg reflector (DBR), anti-reflection layer

## Abstract

Narrow-bandgap germanium–tin (GeSn) is employed to fabricate metal–semiconductor–metal (MSM) near-infrared photodetectors with low-dark currents and high responsivity. To reduce the dark current, the SiO_2_ layer is inserted in between the metal and semiconductor to increase the barrier height, albeit at the expense of photocurrent reduction. To couple more incident light into the absorption layer to enhance the responsivity, the distributed Bragg reflectors (DBRs) are deposited at the bottom of the GeSn substrate while placing the anti-reflection layer on the surface of the absorption layer. With the interdigital electrode spacing and width, both set at 5 µm and with 1 V bias applied, it is found the responsivities of the generic MSM control sample detector, the MSM with DBR, and the MSM with AR layer are 0.644 A/W, 0.716 A/W, and 1.30 A/W, respectively. The corresponding specific detectivities are 8.77 × 10^10^, 1.11 × 10^11^, and 1.77 × 10^11^ cm·Hz^1/2^/W, respectively. The measurement data show that these designs effectively enhance the photocurrent and responsivity. At 1 V bias voltage, normalized responsivity evinces that the photodetection range has been extended from 1550 nm to over 2000 nm, covering the entire telecommunication band. Incorporating GeSn as a sensing layer offers one of the new alternative avenues for IR photodetection.

## 1. Introduction

Infrared (IR) photodetectors are considered a niche in a myriad of applications that involve night vision, astronomy, ballistic missile tracking, microelectronics/nanoelectronics reliability testing, and environmental tracking, just to name a few. Traditionally, infrared radiation can be classified according to a particular wavelength range it falls into. Therefore, in general, a near-wavelength or a short-wavelength IR region corresponds to 0.8 to 3 μm, a range of 3 to 5 μm belongs to a mid-wavelength IR (MWIR) region, while a span of 8 to 12 μm is known as the long-wavelength IR (LWIR) region. To satisfy the particular spectral detection needs, quite a few conventional materials have already been investigated and put into practical use. For example, the aforementioned IR regimes can be conveniently covered by Ge (0.8–1.8 µm), InGaAs (0.9–2.6 µm), PbS (1.0–2.9 µm), PbSe (1.5–4.8 µm), and HgCdTe alloys (2.0–8.0 µm) [1,2]. In addition to these materials, other novel material candidates have also been pursued and studied. For instance, a InN/GaN heterostructural photodetector operating at 1064 nm has been fabricated and evaluated with a responsivity of 37.07 A/W under an illumination intensity of 50 µW/cm^2^ at −1.1 V [3]. Perovskite-based tandem photodetectors incorporating an interlayer of PMMA doped with Au nanoparticles have been tested to yield a responsivity of approximately 381 A/W, under an 850 nm illumination intensity of 270 μW/cm^2^ and at a biasing voltage of 1 V [4]. Other broadband IR photodetectors based on the hybrid bulk heterojunction of PbSe colloidal quantum dots and CsPbBr_1.5_I_1.5_ mixed-halide perovskite nanocrystals have been fabricated and characterized to yield the optical responsivities of 4.46 and 3.87 A/W, under the 980 and 1550 nm illumination intensities of 0.4 and 0.2 mW/cm^2^ at 0 V, respectively [5]. There has also been an attempt to use p-BP/n-PdSe_2_ transition-metal dichalcogenide (TMD) to fabricate a gate-tunable Van der Waals heterostructural photodetector to detect IR radiation with a gate-controlled photoresponsivity of 4.53 × 10^5^ and 1.63 × 10^5^ A/W, under the irradiation of light of IR wavelengths of 1064 and 1310 nm, respectively [6]. However, judging from the unique perspective of the silicon platform, the above-mentioned photodetectors certainly bear very little compatibility with the CMOS-processing flow. Therefore, finding silicon-based semiconducting alloys that are also capable of detecting IR radiation appears to be sensible to pursue.

For more than several decades, silicon and germanium have been the main candidate materials utilized for optoelectronics and photonic integrated circuits [7,8,9]. However, for a detecting spectrum that falls within the infrared regime, silicon photodetectors usually have low responsivity in the short-wave infrared (SWIR) range. In response to this shortcoming, research interest has gradually shifted to Ge because of its direct bandgap that supports a telecommunication wavelength of 1310 nm. Unfortunately, the responsiveness of Ge-based devices drops dramatically beyond 1500 nm [10,11]. Therefore, Ge-based devices are not well-suited for optical detection in the near- and mid-infrared (MIR) range. Germanium is believably a quasi-direct bandgap material with a direct bandgap energy corresponding to a wavelength of 1550 nm. By judiciously alloying Ge with Sn of certain content, the resulting bandgap shrinkage will push the detection wavelength range beyond 2000 nm [11]. Certainly, a wavelength of 2000 nm is not unique for GeSn, since this particular wavelength can also be delivered by conventional III-V compound semiconductor systems [12]. However, for the benefits of silicon photonics and the prospective monolithic integration over silicon, it is vital to mandate the fabrication processes of the photonic components to be complementary metal–oxide–semiconductor (CMOS) compatible. In recent years, GeSn and its relevant alloys have emerged as promising optoelectronic materials for dealing with the aforementioned fabrication and performance concerns [13,14,15,16,17]. It has been reported that as Sn content reaches 3%, the cut-off wavelength of the Ge_1−x_Sn_x_ photodetectors is approximately 1800 nm [18], while increasing the Sn composition x to 0.065 or 6.5% means the resultant 2000 nm detecting band is well-covered [19]. As for the comparison, our 100 nm-thick GeSn film with approximately 8% Sn content is deposited over the 100 nm-thick-Ge-buffered silicon-on-insulator (SOI) substrate using the ultrahigh vacuum chemical vapor deposition technique.

On the other hand, enhancing the responsivity is also one of the vital tasks confronting the designers of optical photodetectors. Several innovative schemes that have been reported up to date include the utilization of photon-trapping microstructures [20], the inclusion of a resonant cavity realized either with the top- and bottom-distributed Bragg reflectors (DBRs) [21] or a buried oxide (BOX) layer as a bottom reflector coupled with a SiO_2_ passive layer as a top reflector [22], and waveguide-coupled photodetection [23]. With a somewhat similar approach but a different operating principle [21], what we propose instead is to introduce the SiO_2_/TiO_2_ dielectric DBR reflector stack along with the SiO_2_ anti-reflection layer to the photodetectors, aiming for the benefit of dialing up the optical responsivity at a near-infrared or a short-wavelength-infrared (SWIR) band. To our knowledge, we believe this is the first time a pure dielectric DBR mirror stack and SiO_2_ antireflection layer have been jointly incorporated with GeSn photodetectors. Unlike other traditional semiconductor-based DBRs such as Si/SiO_2_ and InAs/AlAs, one unparalleled benefit of using the dielectric DBR mirror stacks, such as the SiO_2_/TiO_2_ adopted in the current study, is that its absorption is negligibly small in the visible and near-infrared spectral ranges [24], which translates to the enhanced reflectivity of the stack and tiny absorption-related losses. In terms of contemplating plausible device configurations for photodetection, a metal–semiconductor–metal (MSM) photodetector is selected for the reason of its structural simplicity that comes in the form of back-to-back Schottky diodes, along with the interdigitated metal finger electrodes patterned on a semiconductor layer. To recap what we wish to pursue in this paper, we seek to demonstrate a GeSn MSM photodetector for enhancing the responsivity in the SWIR regime. The objective could be met by equipping the MSM photodetector with a SiO_2_/TiO_2_ DBR stack and SiO_2_ anti-reflection layer structure for benchmark comparison. The final suggestion on how to enhance the detector performance by optimizing the antireflection coating in a form of a multilayer structure is also provided.

## 2. Device Fabrication

A.Metal–semiconductor–metal (MSM) structure

A clean lithography-ready substrate is covered with HMDS beforehand to increase photoresist (PR) adhesion on the substrate surface. The AZ-5214 PR is then coated using a spin coater. To obtain a specific thickness, the PR AZ-5214 is coated at an angular speed of 4000 revolutions per minute (rpm) for 30 s. The PR AZ-5214 coated substrate is then baked at 90 °C for 90 s. After lithographically defining the pattern, 150 nm-thick titanium metal is deposited on the substrate before performing a lift-off step, as depicted in Figure 1a.

B.MSM structure with DBR

The DBR structure is first added to the backside of the specimen chip with six pairs of SiO_2_/TiO_2_ mirror stacks using the electron-beam evaporation technique, and each pair consists of 340 nm-thick SiO_2_ and 180 nm-thick TiO_2_. Both SiO_2_ and TiO_2_ layers are deposited via the electron-beam evaporation technique. Specifically, each SiO_2_ layer is deposited at a rate of 0.7 Å/sec while the chamber pressure is kept at 4.5 × 10^−5^ torrs. For each TiO_2_ layer, the deposition rate is maintained at 0.3 Å/sec while the chamber pressure is also kept at 4.5 × 10^−5^ torrs. The relevant design parameters of the DBR stacks are verified before the actual deposition by carrying out a MATLAB simulation to ensure the reflection spectrum of the mirror stacks coincides with the intended detection wavelength range. The design of our SiO_2_/TiO_2_ reflection stack is based on an incident wavelength of 2000 nm and SiO_2_/TiO_2_ refractive indices of 1.45/2.4 so that the thickness of each layer can be ascertained accordingly.

C.MSM structure with DBR and anti-reflection layer

First, after cleaning the specimen coated with the DBR structure, the SiO_2_ anti-reflection layer is deposited through the electron-beam evaporation method on the surface of the specimen with a thickness of 340 nm, and the pattern is defined by lithography. The anti-reflection is targeted at a wavelength of 2000 nm. Then, the substrate is etched with BOE for 40 s, and titanium metal is deposited on the substrate at 150 nm and lift-off, as depicted in Figure 1b. In the metal electrode section, the sizes of 5, 10, 15, and 20 µm are targeted for the interdigital electrode spacing and width, and the interdigital electrode length is 300 µm, as depicted in Figure 2.

## 3. Results and Discussion

A.Metal–semiconductor–metal (MSM) structure

The MSM photodetector performance is investigated in terms of measuring the dark current and the photocurrent. The dark current is measured under no illumination, and Figure 3a shows the comparison of the dark current-voltage characteristics measured for samples of four different sizes (5, 10, 15, and 20 µm). The photocurrent is quantified by using a light source with a wavelength of 2000 nm to irradiate the surface of the photodetector. Figure 3b shows the comparison of photocurrent-to-voltage relations recorded for the four samples, as mentioned previously.

By ranking the magnitude of the current from the highest to the lowest, the smallest electrode size of 5 µm yields the highest current while the largest size of 20 µm has the lowest current. This observation is attributed to the fact that as the distance between the interdigital electrode spacing increases, the electron-hole pairs (EHPs) distancing away from the respective electrodes also increase, thereby augmenting the resistance they encounter or decreasing the corresponding current at the same voltage. Figure 4 shows the comparison of photocurrent (*I_photo_*) and dark current (*I_dark_*) between samples of different sizes. The photocurrent surpassing the dark current typically represents the generation of the EHPs instigated by the incident light source, followed by a subsequent drift to electrodes due to the applied voltage.

The dark current can be curtailed by adding a 10 nm-thick SiO_2_ layer between the interdigital electrodes and GeSn, turning it into a metal–insulator–semiconductor (MIS)-based photodetector. By comparing with a conventional MSM device sample, the dark current reduction has been consistently observed from the MIS-based photodetectors of all different electrode sizes. For instance, the dark current of an MIS-based photodetector covered with interdigital electrodes of 5 μm finger/spacing widths can be brought down by more than an order of magnitude, as shown in Figure 5a. However, its photocurrent, unfortunately, has also been compromised by less than an order of magnitude, as depicted in Figure 5b. What we have observed can be properly elucidated by using a set of schematic band diagrams presented in Figure 6. As clearly depicted in Figure 6a, inserting a thin SiO_2_ layer helps to elevate the barrier height between metal and GeSn, which prevents the interfacially and bulk-trapped carriers from transporting into metal after being excited into the conduction band. However, as depicted in Figure 6b, the photo carriers generated after light illumination have also been impeded from crossing over the barrier into metal, thereby bringing down the magnitude of photocurrent.

The entire measurement setup consists of a probe station, an IR light source for wavelengths up to 2100 nm, a source meter, and data acquisition software. Specifically, the current-voltage measurement data of our photodetectors are collected with a Keithley 2632A source meter. The light source used to deliver light of different wavelengths needed for the responsivity and specific detectivity characterizations is achieved by using a Newport 250 W quartz–tungsten–halogen research light source connected with a monochromator.

The responsivity is directly proportional to the level of photocurrent detected. The responsivity *R* can be determined by *R* = (*I_photo_* − *I_dark_*)/*P_in_*, where *P_in_* is the output power incident on the GeSn absorption layer. Since metal blocks the incident photons from reaching the GeSn absorption layer, the responsivity is extracted by excluding the shadowing effect of the metal contacts on the top of the GeSn MSM PDs. As can be seen in Figure 7, the responsivity is estimated to be 0.644 A/W, 0.133 A/W, 26.5 mA/W, and 21.2 mA/W for the size of 5, 10, 15, and 20 µm, at an incident wavelength of 2000 nm at 1 V bias voltage, respectively. Furthermore, the specific detectivity (*D**) can be calculated in terms of the responsivity *R* as $D^{*}=R\sqrt{A/2qI_{dark}}$, where *A* is the effective area for light detection and *q* is the electronic charge. Therefore, the highest responsivity achieved for 5 μm interdigital spacing, 0.644 A/W, would lead to the specific detectivity *D** of 8.77 × 10^10^ cm·Hz^1/2^/W.

B.MSM structure with DBR

In this section, we propose ways to increase the responsivity and optimize its structure to achieve the highest performance at the absorption wavelength of 2000 nm. Figure 8 shows the reflectance at 2000 nm for different SiO_2_/TiO_2_ pairs (N) calculated using the transfer matrix method (TMM). Due to the large difference in the refractive index between SiO_2_ and TiO_2_, high reflectivity is expected. As N increases, the reflectance increases significantly and approaches a theoretical value of 1 for N larger than 4 within the 1600–2600 nm spectral range. Therefore, as the figure manifests, the sample coated with four pairs of DBRs yields nearly 100% reflectivity. Figure 8(b) shows the measured reflectivity, and it is found that six pairs of DBRs would have the best reflectance performance closest to 100%.

Figure 9 shows the relationship between the measured photocurrent and the voltage at an incident wavelength of 2000 nm for samples with four different sizes of interdigital electrode spacing and width coupled with the DBR structure. A similar trend among samples of different electrode dimensions has also been observed. Figure 10 shows the comparison of photocurrent and dark current between different sizes with DBR structures. The data indicate that at least an order of magnitude difference in the current contrasts has been achieved for all the samples under study.

As can be seen in Figure 11, the responsivity is estimated to be 0.716 A/W, 0.307 A/W, 53 mA/W, and 27.71 mA/W for the size of 5, 10, 15, and 20 µm, at a wavelength of 2000 nm at 1 V bias voltage, respectively. Accordingly, the photodetector with 5 μm electrode linewidth/spacing consistently outperforms the other devices with other dimensions. Overall, the photodetectors with the DBRs, regardless of their electrode sizes, have all brought forth an increase in photocurrent. Consequently, the highest responsivity of 0.716 A/W translates to the specific detectivity *D** of 1.11 × 10^11^ cm·Hz^1/2^/W. Figure 12 specifically shows the comparison of the photocurrents before and after the inclusion of six pairs of DBR. Adding the DBR to the GeSn photodetector effectively increases the photocurrent, and the responsivity becomes significantly higher as a result.

C.MSM structure with DBR and anti-reflection layer

One of the loss-related issues concerning the operation of GeSn in a MSM photodetector is the reflection of light from its surface. GeSn has a high index of refraction (*n* = 4.3), and therefore the Ge surface tends to reflect more than 30 to 40% of the incident light. As a result, the reduction of surface reflectance is essential for enhancing the performance of the GeSn-based infrared photodetectors. It is well known that an optimized anti-reflection (AR) coating is required to ascertain a substantial amount of incident light that could reach and penetrate GeSn film for absorption. Among the possible materials considered, SiO_2_ has been considered a superior AR candidate for the GeSn-based photodetector because of its good passivation, scratch resistance, and chemical stability. This is confirmed by the reflection spectra collected in the wavelength range of 1000~2300 nm for GeSn substrates with and without the incorporation of a SiO_2_ AR layer of 340 nm in thickness, as shown in Figure 13.

To investigate the beneficial effect of the SiO_2_ AR layer on the optoelectronic performance of the GeSn MSM photodetectors, a series of photoelectric measurements are performed at room temperature. Figure 14 shows the relationship between the measured photocurrent and voltage at an incident wavelength of 2000 nm for detectors with four different sizes of interdigital electrode spacing and width and also with the AR layer and DBR structure properly coated. Figure 15 shows the comparison of photocurrent and dark current obtained from samples of different electrode sizes with both the AR layer and DBR structure incorporated. Accordingly, the photocurrent/dark current contrast ratios are found to be approximately two orders of magnitude for all samples under study. Furthermore, as can be seen in Figure 16, the responsivity is estimated to be 1.30 A/W, 0.851 A/W, 95.7 mA/W, and 27.63 mA/W for the samples with the electrode size of 5, 10, 15, and 20 µm at the light wavelength of 2000 nm and 1 V bias voltage, respectively. Furthermore, it can also be found that the highest responsivity of 1.30 A/W corresponds to the specific detectivity *D** of 1.77 × 10^11^ cm·Hz^1/2^/W.

To determine whether adding the AR layer to the photodetector would lead to an increase in photocurrent, the experimental result compiled in Figure 17 shows the photocurrents obtained from devices with and without the inclusion of the AR layer. For MSM photodetector with the electrode size of 5 and 10 µm, adding the AR layer effectively increases the photocurrent, and this in turn renders the responsivity significantly higher. On the other hand, for electrode sizes of 15 and 20 µm, adding the AR layer may not be beneficial for enhancing the magnitude of the photocurrent. We believe that there are two reasons for this observation: first, the possibility of surface defects situated in GeSn would make the photo carriers generated during illumination at risk of getting intercepted by defects before they have the opportunity to reach the electrodes. Second, when the distance between the electrode spacing is large, though the illumination area potentially becomes larger, a longer transit time for these carriers relative to their corresponding lifetime is conceivably needed, which in turn raises the probability of capturing these carriers by defects embedded in the film.

Furthermore, there is certainly room for improving the performance of AR coating. Recall that our SiO_2_ is designed for antireflection at a 2000 nm wavelength. However, due to the stoichiometric inhomogeneity of the SiO_2_ layer deposited with electron-beam evaporation that leads to index variation, this AR layer may deviate away from its intended performance at a wavelength of 2000 nm. Moreover, for a broadband photodetector application, it would be preferable to extend the ultralow reflectance to a wider bandwidth; in this case, the number of quarter-wavelength layers needed may have to be two or more. Since GeSn is a high-index material with a refractive index of more than 4.0 and could potentially lead to more than 40% of the reflection loss, for a case that involves a double-layer AR coating deposited on the GeSn layer with a layer structure of GeSn(*n_s_*)/*H*(*n*_2_)/*L*(*n_1_*)/air(*n*_0_) and with each layer a quarter-wave thick—where *n_s_* > *n*_2_ > *n_1_* > *n*_0_—the reflectance can be cut down to zero at a comparatively wider bandwidth by satisfying the relation: *n*_1_/*n*_0_ = *n*_2_/*n*_1_ = *n_s_*/*n*_2_. To extend this zero-reflectance bandwidth even further, a multilayer coating would be needed, such as a three-layer coating with each layer a quarter-wave thick, and the relation of *n*_1_/*n*_0_ = *n*_2_/*n*_1_ = *n*_3_/*n*_2_ = *n_s_*/*n*_3_ is required to be fulfilled. By taking the aforementioned improvement into account, we expect the responsivity and specific detectivity of the GeSn-based infrared photodetectors can be enhanced dramatically. Therefore, without the benefits of the elaborating or fancy GeSn layer structures, such as quantum wells embedded within a resonant cavity, a simple GeSn layer with decent material quality can still perform marvelously with the addition of a DBR mirror stack and the optimized antireflection layer.

The normalized spectral responsivity is displayed in Figure 18. The photodetector exhibits responsivity at NIR (1800 to 2100 nm) with a sharp downfall beyond 1950–2000 nm, which is correlated to the absorption edge and bandgap of GeSn. Since the cutoff wavelength of Ge falls between 1550 and 1800 nm, alloying 8% Sn with Ge extends the absorption of our Ge_0.94_Sn_0.06_ samples to the near NIR or close to 2000 nm wavelength, which has been empirically confirmed by the spectral responsivity measurement. To compare our results with other research teams’ published works based on the criteria of the metal-semiconductor-metal (MSM) and single GeSn absorption layer configurations, the 9% Sn-based MSM photodetectors fabricated by Son et al. earlier have managed to deliver the responsivities of 0.5 and 0.29 A/W at a wavelength of 1600 and 2033 nm at 2 V, respectively [25]. It appears that the highest responsivity (~1.3 A/W) achievable through our 8% Sn-based device with DBR and AR both included has an edge at a wavelength of 2000 nm and a bias voltage of 1 V. Furthermore, Ghosh et al. have realized a wide bandwidth infrared detector with 4% Sn composition to cover a wavelength range of 1200 to 2100 nm [26]. The responsivity of their detector (in the mA/W level) peaks at the wavelength of 1200 nm and gradually dials down, and eventually levels off within a wavelength range of 1800 to 2100 nm when administering a bias voltage of 3 to 7 V. At 1 V, which is the voltage we have consistently used to characterize our detector’s responsivity at 2000 nm wavelength, their resultant responsivity measured is barely noticeable, which is well explainable because of their low Sn composition (4%). Therefore, based on the aforementioned results we have achieved so far, the GeSn alloy certainly merits classification as a suitable material for applications in NIR or SWIR photodetection in its own right.

## 4. Conclusions

In summary, incorporating the DBR structure with the MSM photodetector enhances both the photocurrent and the responsivity. The addition of a 340 nm SiO_2_ AR layer is found to be effective in reducing the surface reflectance of GeSn, allowing more light to enter the absorber layer for the benefit of raising the photocurrent and responsivity. Taking the electrode size of 5 µm as an example, the responsivity of a purely MSM structure is 0.644 A/W, and the corresponding specific detectivity is 8.77 × 10^10^ cm·Hz^1/2^/W. With an additional DBR structure included, the responsivity increases to 0.716 A/W while its specific detectivity elevates to 1.11 × 10^11^ cm·Hz^1/2^/W. Then, after introducing an extra AR layer, the responsivity reaches up to 1.30 A/W, which leads to the specific detectivity of 1.77 × 10^11^ cm·Hz^1/2^/W. The results show that combining the DBR and AR layer with an MSM could effectively improve the performance of the SWIR photodetector.

Finally, the normalized responsivity measurements show that the cutoff wavelength can be extended to near 2000 nm, which exceeds the highest absorption wavelength limit of Ge, thereby extending the traditional communication band to the short-wavelength infrared (SWIR) regime and ultimately rendering GeSn as one of the leading materials for fabricating SWIR photodetectors.

## Figures and Tables

**Figure 1 micromachines-13-01733-f001:**
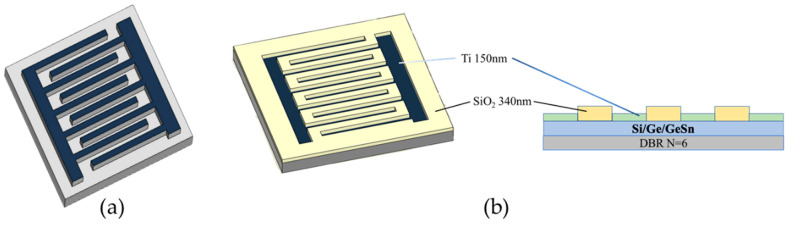
(**a**) Top view of MSM photodetector design. (**b**) The bird’s eye and cross-sectional views of the MSM photodetector with DBR and anti-reflection layer.

**Figure 2 micromachines-13-01733-f002:**
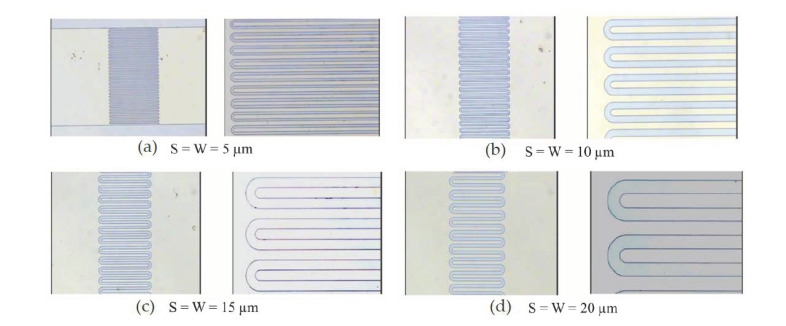
The sizes are (**a**) 5, (**b**) 10, (**c**) 15, and (**d**) 20 µm for the interdigital electrode spacing and width.

**Figure 3 micromachines-13-01733-f003:**
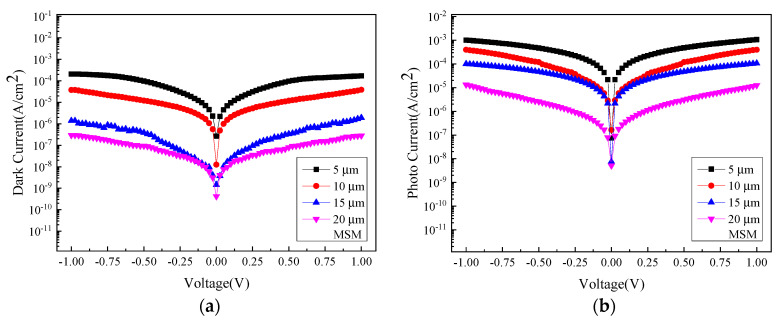
The current density–voltage relations for the MSM photodetectors with four different sizes: (**a**) under no illumination, and (**b**) under the illumination of 2000 nm wavelength.

**Figure 4 micromachines-13-01733-f004:**
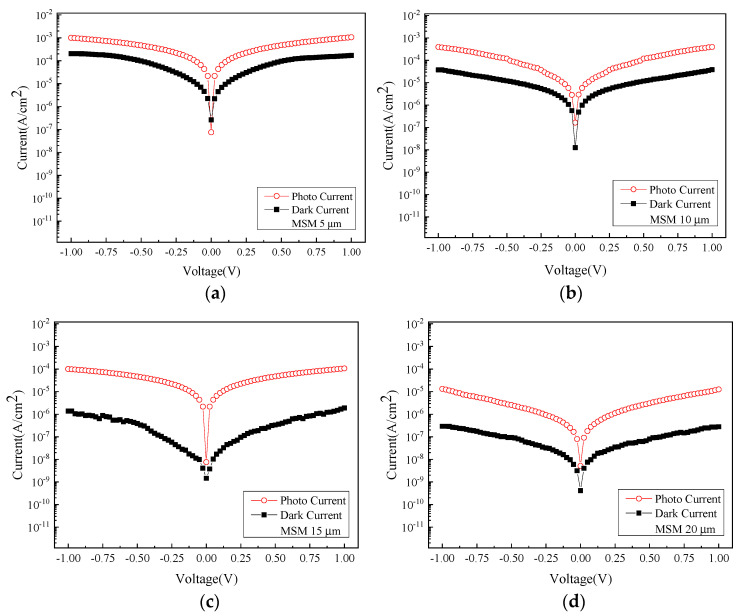
The photocurrent and dark current density between different sizes: (**a**) 5 µm, (**b**) 10 µm, (**c**) 15 µm, and (**d**) 20 µm.

**Figure 5 micromachines-13-01733-f005:**
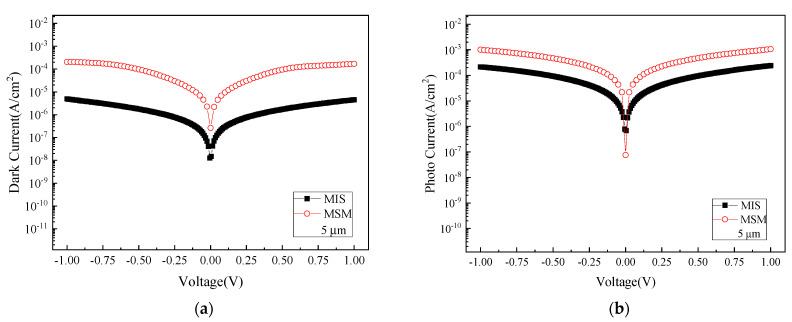
The current-voltage comparison between the MIS-based and conventional MSM photodetectors: (**a**) the dark current and (**b**) photocurrent.

**Figure 6 micromachines-13-01733-f006:**
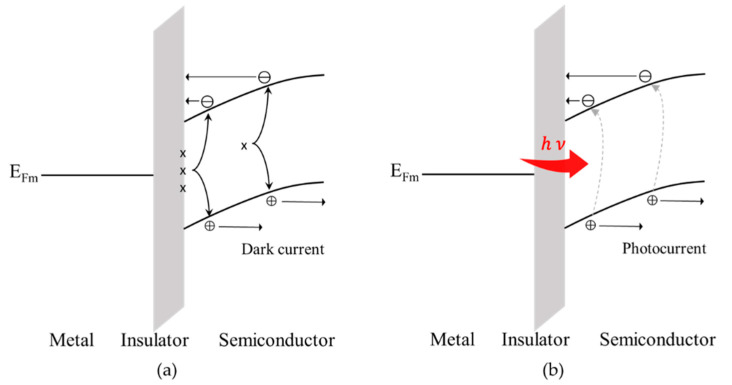
Band diagrams of an MIS diode: (**a**) in a dark condition and (**b**) under light illumination.

**Figure 7 micromachines-13-01733-f007:**
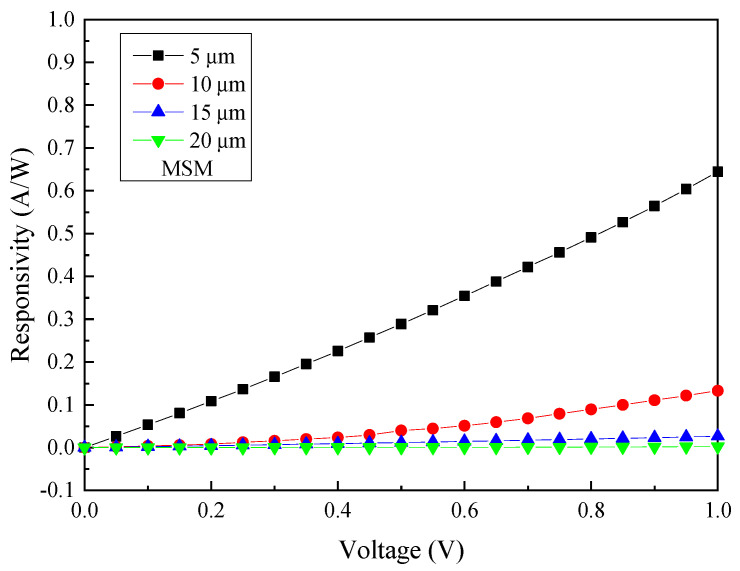
The responsivity of GeSn MSM PD at different four sizes under 2000 nm illumination.

**Figure 8 micromachines-13-01733-f008:**
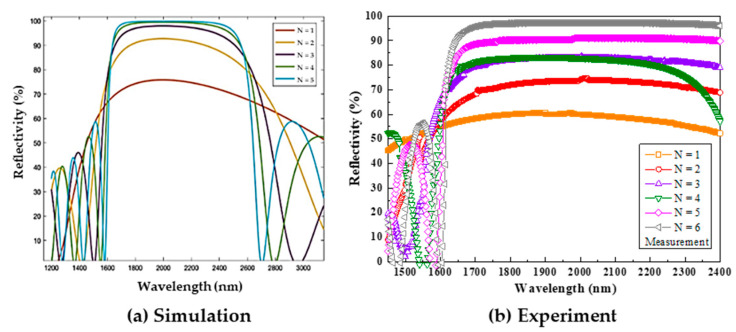
(**a**) Simulated and (**b**) measured reflectivity spectra of the SiO_2_/TiO_2_ DBR with the corresponding thicknesses of 340 nm/180 nm.

**Figure 9 micromachines-13-01733-f009:**
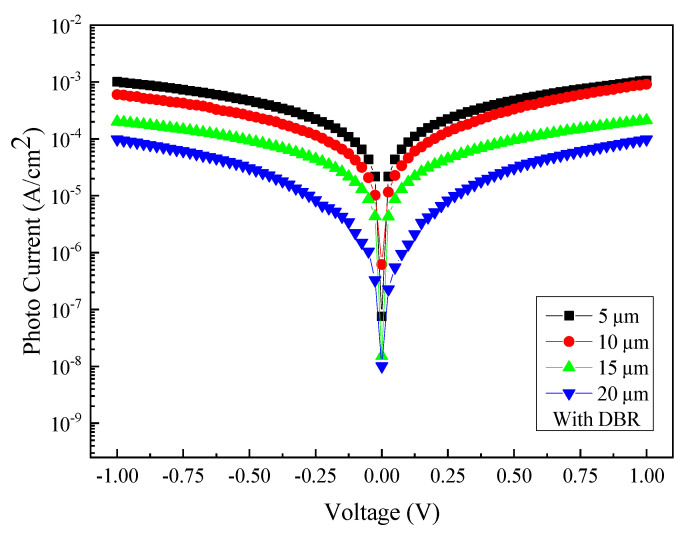
The photocurrent density of PDs with electrodes of different sizes (5 µm, 10 µm, 15 µm, and 20 µm) and with DBR structures coated.

**Figure 10 micromachines-13-01733-f010:**
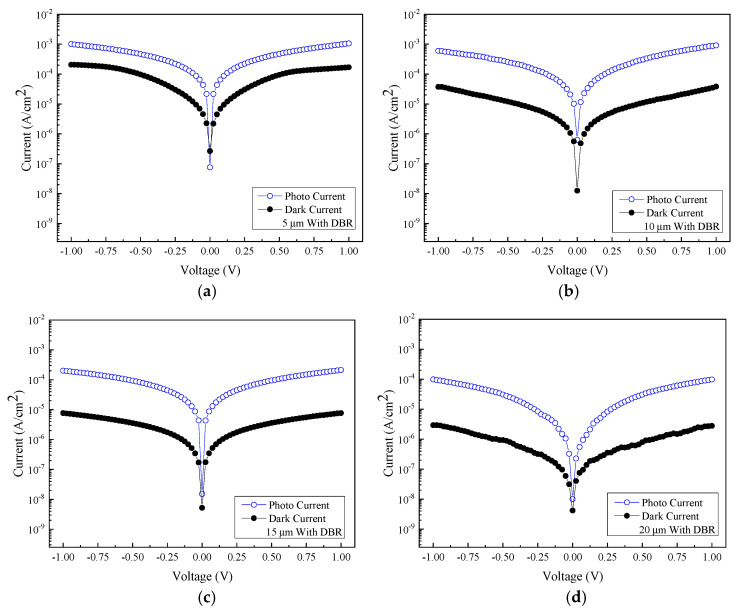
The photocurrent density of the DBR-coated MSM photodetector with four different sizes under the illumination of 2000 nm: (**a**) 5 µm, (**b**) 10 µm, (**c**) 15 µm, and (**d**) 20 µm.

**Figure 11 micromachines-13-01733-f011:**
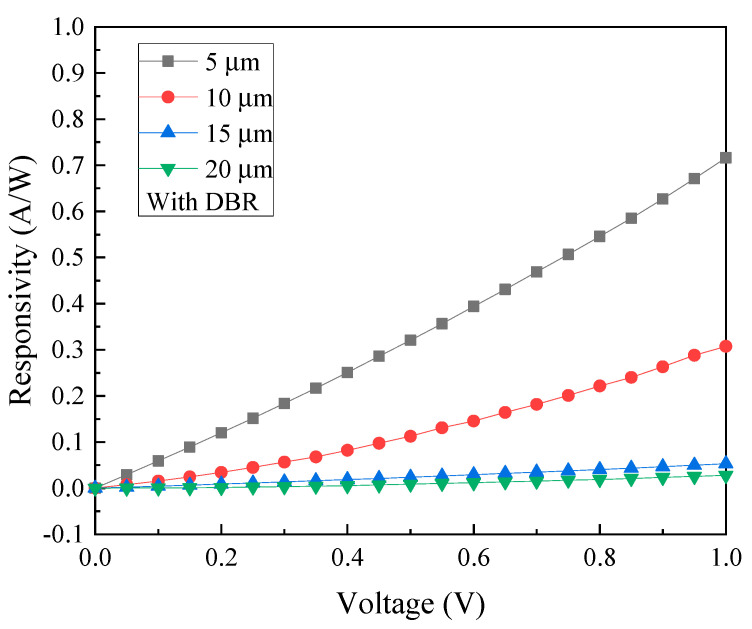
The responsivity of GeSn MSM PD with DBR structure and four different electrode sizes under 2000 nm illumination.

**Figure 12 micromachines-13-01733-f012:**
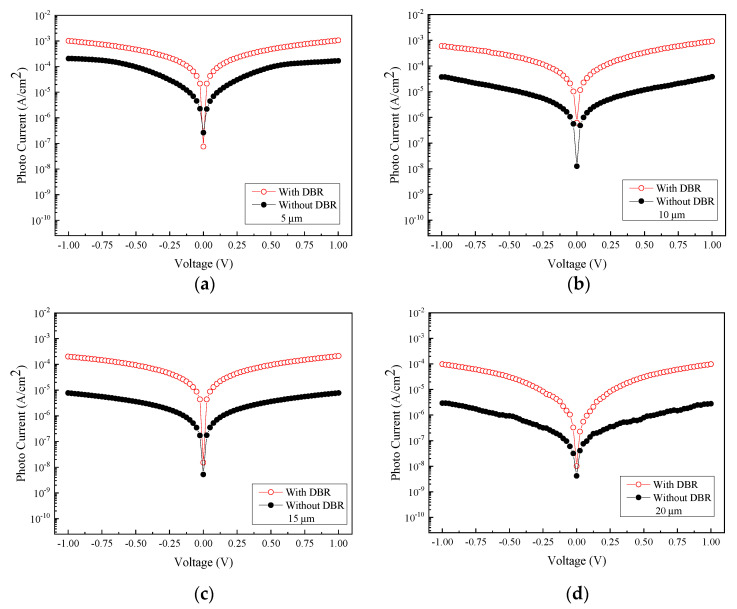
Comparison of photocurrents before and after introducing DBR to the PDs of four different electrode sizes: (**a**) 5 µm, (**b**) 10 µm, (**c**) 15 µm, and (**d**) 20 µm.

**Figure 13 micromachines-13-01733-f013:**
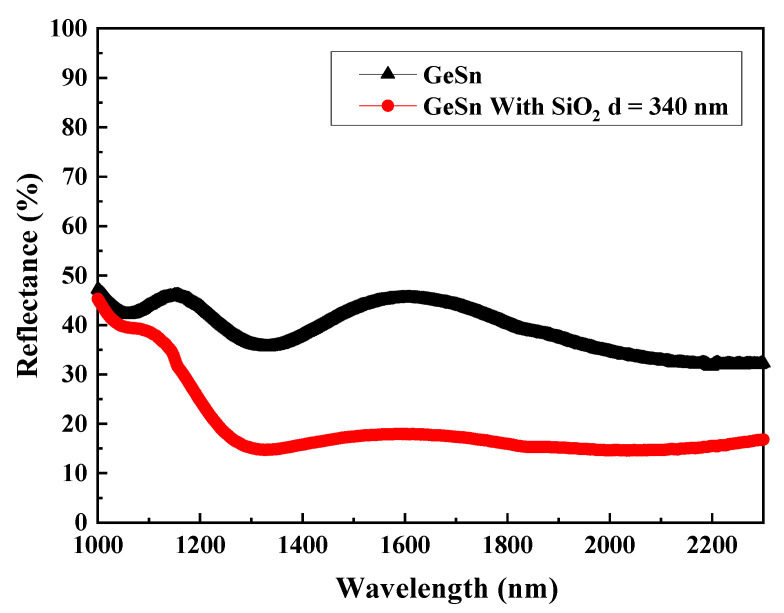
The reflectance plots of the GeSn samples with and without the coverage of the SiO_2_ AR layers of 340 nm in thickness as a function of wavelength in the range of 1000~2300 nm.

**Figure 14 micromachines-13-01733-f014:**
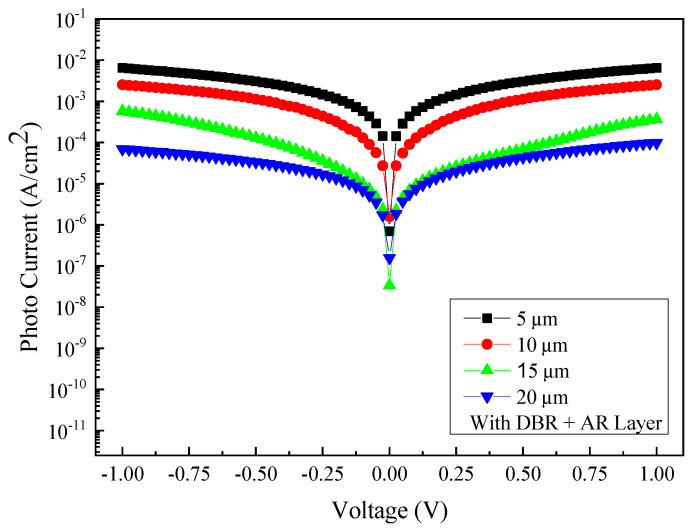
The photocurrent current of the AR/DBR-coated PDs with electrodes of four different sizes: 5 µm, 10 µm, 15 µm, and 20 µm.

**Figure 15 micromachines-13-01733-f015:**
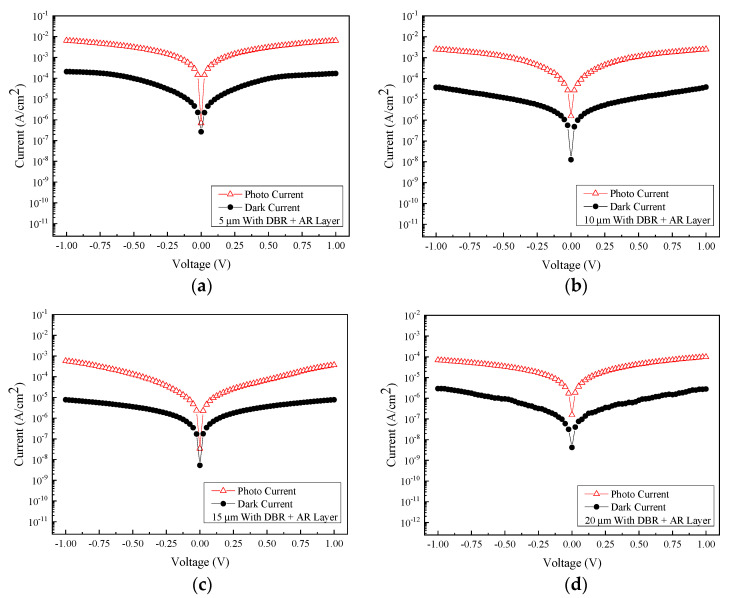
The respective photocurrent and dark current are recorded from the MSM PDs with four different electrode sizes with and without the illumination of 2000 nm light. All samples are coated with DBR mirror stacks: (**a**) 5 μm, (**b**) 10 μm, (**c**) 15 μm, and (**d**) 20 μm.

**Figure 16 micromachines-13-01733-f016:**
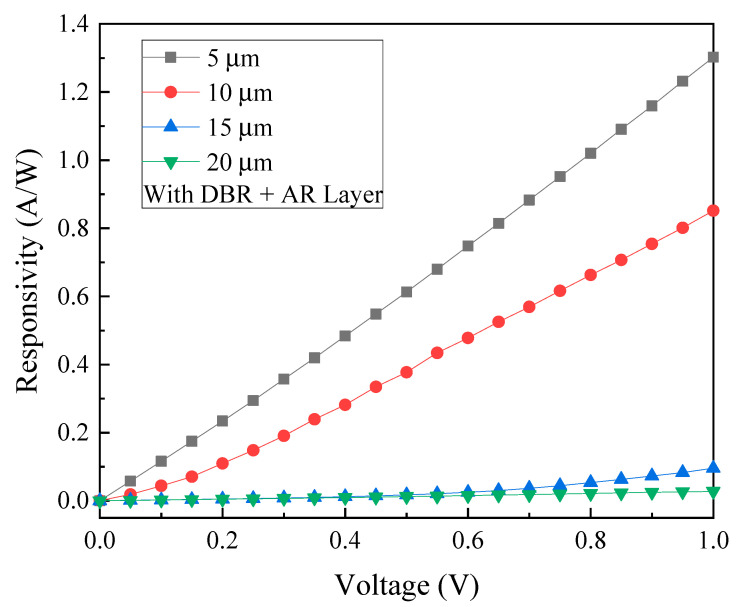
The responsivity of GeSn MSM PDs with combined AR layer and DBR structure, and with electrodes of four different sizes under 2000 nm illumination.

**Figure 17 micromachines-13-01733-f017:**
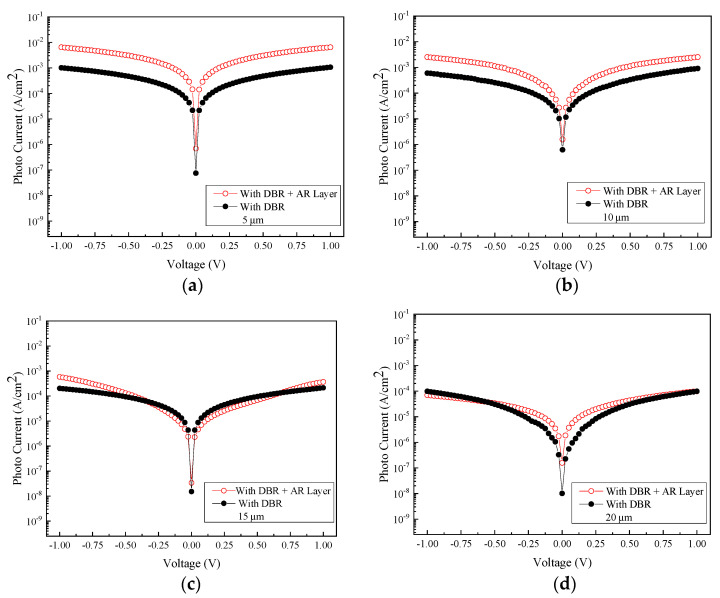
Comparison of photocurrents recorded before and after introducing the AR layers to the samples of different electrode sizes: (**a**) 5 μm, (**b**) 10 μm, (**c**) 15 μm, and (**d**) 20 μm.

**Figure 18 micromachines-13-01733-f018:**
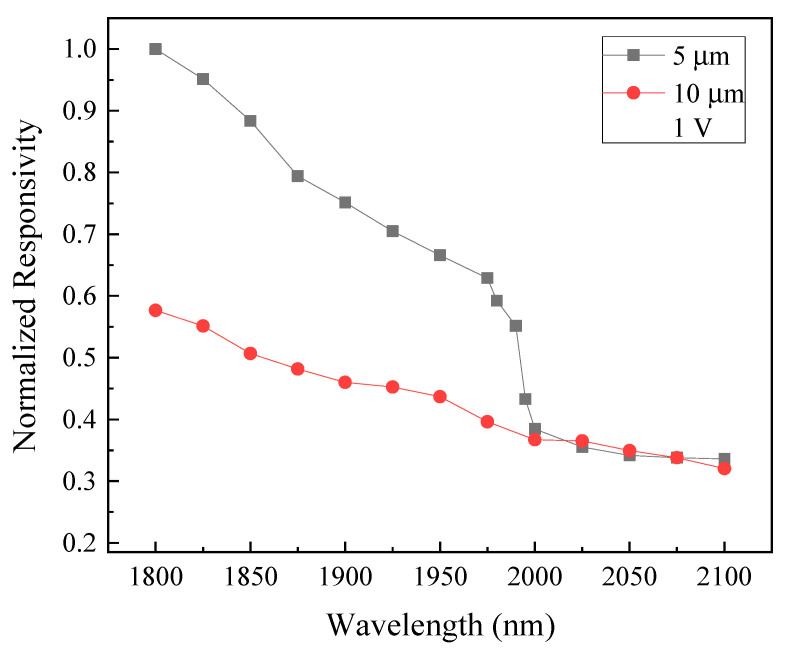
The normalized spectral responsivity of the GeSn MSM photodetector with AR layer and DBR structure incorporated.

## Data Availability

Data is contained within the article.

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
