# Peer review of "Germanium-Tin (GeSn) Metal-Semiconductor-Metal (MSM) Near-Infrared Photodetectors"

_micromachines, 2022, doi:10.3390/mi13101733_

Round 1

Reviewer 2 Report

Recommendation: Major revision

In this manuscript, the authors reported the fabrication of GeSn metal-semiconductor- metal near-infrared photodetectors with low-dark current and high responsivity. An initial increase in the responsivity was observed at low electrode size. To enhance the responsivity, the distributed Bragg reflectors are deposited at the bottom of the GeSn substrate while placing the anti-reflection layer on the surface of the absorption layer. The results show that combining the DBR and AR layer with an MSM effectively improves the performance of the photodetector. In my opinion, the present manuscript lacks in many aspects and calculations perspective. Therefore, I can only recommend it for publication in the Journal of Micromachines after the authors have addressed the following major issues:

1. Details of the growth of the GeSn layer, DBR, and AR layer should be expressed in the text.

2. Schematic diagram of the GeSn metal-semiconductor-metal with and without DBR and AR layer is missing. The side view of optimized structures along with the dimensions of the different regions should be presented.

3. The authors didn't present adequate information about the morphology, and composition of the GeSn layer. Characterization includes cross-sectional transmission electron microscopy (XTEM), and SEM images are necessary. The Sn concentration and strain of the GeSn layer should be characterized. Especially, photoluminescence (PL) measurement results are necessary to confirm the band gap of the grown GeSn.

4. In the manuscript, details of the electrical and optical measuring instruments are missing. Add information about measurement instruments.

5. The responsivity calculation equation missing and should show.

6. In the abstract, the authors write: " To reduce the dark current, the SiO2 layer is inserted in between metal and semiconductor to increase the barrier height, albeit at the expense of photocurrent reduction." The authors should show the schematic band diagram and discuss this in-depth.

7. How are the performances here compared with state-of-the-art reports, especially MSM structures-based photodetectors? The readers would like to see a paragraph near the end of the manuscript before the conclusion dedicated to such a comparison. More recent literature is suggested to be included as a comparison.

8. The English should be polished by a native speaker. There are several typos and grammatical errors throughout the manuscript such as inserted in between; 6% of Sn; in a form of; comes significantly higher as a result; Furthermore, As can be seen in Figure 14 etc.

9. I also suggest authors refer to some related publications such as Micromachines 11, no. 9 (2020): 795. https://doi.org/10.3390/mi11090795; IEEE Sensors Journal 21, no. 8 (2021): 9900-9908. https://doi.org/10.1109/JSEN.2021.3054475; Optical Components and Materials XVI, vol. 10914, pp. 244-250. SPIE, 2019. https://doi.org/10.1117/12.2506355.

Reviewer 3 Report

Manuscript ID: micromachines-1944039 

The manuscript entitled “Germanium-Tin (GeSn) Metal-Semiconductor-Metal (MSM) Near-infrared Photodetectors” is presented by R. W Chuang in which they described the fabrication of MSM near-IR photodetector and tried to reduce the dark current by inserting the SiO2 layer between metal and semiconductor. The manuscript is normally written and presented results are somehow interesting. Therefore, it can be accepted after some modification

Recommendation: Revision.

1.      Author should add some novelty and motivation of the work in introduction section which should be attractive for the readers.

2.      The specific detectivity of the device should also be measured and mention in results section as well as abstract and conclusion sections.

3.      There are multiple figures presented in the results section. Author should merge/combine some of them to reduce the spacing and add some of them into sporting information. For example, Figure 1, 3 and Figure 4 should be combined together. Figure 2 should be shifted to the sporting information and so on.

4.      Figure 5, 6 and 7 can be combed. Figure 4, 8, 10, 13 and 15 can be merged into (a and b) instead of (a, b, c, and d) etc.

5.      Although author has added number of results but this manuscript is still suffering from lack of innovation, inspiration and literature survey.

6.      Author should also add the device physics of photodetectors and carrier transportation mechanisms within the device. I’m suggesting few references for the help to add device physics for photodetectors and carrier transportations. Author should consider them carefully. 10.1002/adfm.202201527; 10.1002/adma.202002628; 10.1016/j.mtphys.2022.100829; 10.1016/j.surfin.2022.101772; 10.1002/admi.202200017; 10.1002/admi.202200105; 10.1007/s40843-021-1895-2; 10.1021/acsami.0c04093; 10.1021/acsami.9b22898; 10.1016/j.sna.2019.07.003.

Round 2

Reviewer 2 Report

The author(s) completed our notes and suggestions in the revised manuscript. Therefore, the revised manuscript is ready for publication.

Reviewer 3 Report

Revision is satisfactory